Knowledge and competence in vestibular rehabilitation: a cross-sectional study of physical therapy interns

Alyahya Danah 1
Alharbi Arwa 1
Kashoo Faizan f.kashoo@mu.edu.sa 1
Alsaheli Shaikha 1
AlMubarak Faisal Mubarak 2
Aljuhni Rehab 1
1 Department of Physical Therapy and Rehabilitation Sciences, Applied Medical Sciences College, Majmaah University , Al Majmaah , Saudi Arabia
2 Physical Therapy, Ministry of Health , Abha , Saudi Arabia
Anson Lesley
Electronic publication date: 2025 Nov 4
Publication date: 2025
Volume: 13
Electronic Location ID: e20213
Received 2025 Feb 20; Accepted 2025 Sep 19
Copyright: ©2025 Alyahya et al.
Copyright year: 2025
Copyright holder: Alyahya et al.
License: This is an open access article distributed under the terms of the Creative Commons Attribution License, which permits unrestricted use, distribution, reproduction and adaptation in any medium and for any purpose provided that it is properly attributed. For attribution, the original author(s), title, publication source (PeerJ) and either DOI or URL of the article must be cited.
License URL: https://creativecommons.org/licenses/by/4.0/

Keywords: Vestibular rehabilitation, Physical therapy interns, Knowledge discrepancy, Clinical competence, Learning methods, Objective assessment, Training experiences

Funding: The Deanship of Postgraduate Studies and Scientific Research at Majmaah University through project number R-2025-2016 Funding was provided by the Deanship of Postgraduate Studies and Scientific Research at Majmaah University through project number R-2025-2016. The funders had no role in study design, data collection and analysis, decision to publish, or preparation of the manuscript.

==============================
Background

Vestibular rehabilitation (VR) is a specialized physical therapy practice area that requires comprehensive knowledge and clinical competence to manage vestibular disorders effectively, which significantly impacts patients’ quality of life. Discrepancies between knowledge and competence among physical therapy interns remain under-researched. Addressing this gap is essential for identifying specific educational deficiencies, improving training programs, and enhancing clinical preparedness.

Objective

This study aimed to examine the knowledge and competence of Saudi physical therapist (PT) interns in VR.

Methods

A cross-sectional survey was conducted among PT interns (n = 233), and VR knowledge was assessed through self-reports and objective testing. The questionnaire, developed via the Delphi method, included items specifically designed to assess competence in and knowledge of VR. Statistical analyses included descriptive statistics, multiple response analysis, and multiple linear regression to explore demographic data, knowledge, and competence predictors. The data were analyzed via JASP 18.1.1.

Results

Competence showed a high mean score of 5.35 out of 8 (67%). In contrast, case report knowledge was low at 0.58/2 (29%), VR tests learned during academic years averaged 4.17 out of 20 (28%), while those learned during the internship averaged 2.43 out of 20 (17%). Similarly, treatment approaches and maneuvers for vestibular-related disorders learned during academic years averaged 2.47 out of 12 (25%), while those learned during the internship averaged 1.69 out of 12 (19%), indicating limited knowledge and skills in vestibular rehabilitation across both academic and clinical training phases. A simultaneous-entry multiple regression indicated that the predictor set significantly explained competence, F(12, 219) = 5.96, p < .001, accounting for 24.6% of the variance (adjusted R2 = .21, RMSE = 1.78). Internship vestibular-test knowledge (B = 0.25, β = .32, p < .001) and workshop attendance (B = 0.82, p = .006) were the only unique contributors. A companion model for case-report knowledge was also significant, F(12, 219) = 2.50, p = .004, but modest (adjusted R2 = .07, RMSE = 0.74); workshop attendance had a positive effect (B = 0.49, p < .001) while possession of a professional physical-therapy degree predicted lower scores (B =  − 1.06, p = .045), with all other variables non-significant.

Conclusion

The findings of this study underscore the need to implement structured academic education in VR. Integrating formal education with hands-on experiences in VR curricula could enhance knowledge and competence among PT interns.

Introduction

Physical therapy (PT) education programs in Saudi Arabia began in the mid-1980s, with King Saud University establishing the first bachelor’s program in PT (Alghadir et al., 2015). Since then, the number of PT schools has steadily increased, highlighting the growing demand for PT training programs across Saudi Arabia (Chahal et al., 2019). PT is a rapidly expanding profession focused on health promotion, disease prevention, and evidence-based treatment techniques to manage symptoms, restore function, and improve quality of life. PT education programs offer various specialties that enable practitioners to treat patients with neurological, cardiovascular, orthopedic, and pulmonary conditions (Khalid, Malik & Khan, 2013; Tawiah et al., 2021). Among these diverse areas of specialization, vestibular rehabilitation (VR) has emerged as a distinct and growing field within neurological physical therapy. The American Physical Therapy Association (APTA) recognized VR as a specialized area within neurology (Hall et al., 2016; Van de Berg et al., 2022). Furthermore, the Barany Society Ad Hoc Committee on Vestibular Rehabilitation recommended that VR be incorporated into PT education (Cohen et al., 2009). This recognition highlights the growing importance of VR as a core component of physical therapy practice, necessitating a solid understanding of its principles and applications.

Knowledge of VR is essential for managing symptoms such as vertigo, nausea, vomiting, intolerance to head motion, nystagmus, unsteady gait, and postural instability, and for the early diagnosis and treatment of vestibular disorders including vestibular hypofunction, benign paroxysmal positional vertigo (BPPV), vestibular neuritis, traumatic brain injury, postural-perceptual dizziness, and vestibular migraine (Porciuncula, Johnson & Glickman, 2012; Alghadir & Anwer, 2018; Dlugaiczyk et al., 2018; Kleffelgaard et al., 2019; Sulway & Whitney, 2019; Meldrum et al., 2020; Huang et al., 2024; Azeez & Nada, 2025). Studies suggest early referrals for VR not only help manage symptoms but also improve postural control, balance, and mobility; reduce anxiety; and optimize quality of life (Mira, 2008; Anderson et al., 2025; Behr, Massa & Honaker, 2025). VR therapeutic techniques include vestibular system stimulation and central compensation through head and trunk movements, which are based on central neuroplasticity mechanisms such as adaptation, habituation, substitution, and BPPV treatment maneuvers (Roy et al., 2024).

In VR, three types of exercises are used to promote central compensation and reduce symptoms such as dizziness and imbalance: habituation, adaptation, and substitution. Habituation exercises (reduce motion sensitivity) involve repetitive movements or provocative stimuli to desensitize the vestibular system. For example, Brandt-Daroff exercises which involve repeated head turns or positional changes that provoke mild dizziness, causing desensitization of the vestibular system (Alashram, 2024). Adaptation exercises (improve vestibulo-ocular reflex and gaze stability) involve a series of head and eye movements designed to help the central nervous system adjust to changes or losses in vestibular input (Rinaudo et al., 2021). Finally, substitution exercises (compensate via other systems) involve the use of vestibular signals in combination with visual and somatosensory cues to enhance central programming, improve gaze stability, and promote postural stability (Sharma & Gupta, 2020).

Developing knowledge and skills both during the academic years and throughout the internship is crucial, as it lays the foundation for effective clinical practice in the future (Korpi, Piirainen & Peltokallio, 2017). A strong theoretical knowledge acquired during academic training, when reinforced and applied during internship, enables interns to confidently manage real-world clinical cases. This integrated learning approach ensures that future physiotherapists are well-prepared to deliver safe, evidence-based vestibular rehabilitation, ultimately leading to improved patient outcomes and enhanced quality of care. We hypothesize that physiotherapy interns who have completed a formal vestibular-rehabilitation curriculum and workshops will achieve higher knowledge and competence scores than those without such exposure. Thus, this study aims to understand the knowledge and competence of PT interns in VR techniques and maneuvers in Saudi Arabia.

Materials & Methods

A cross-sectional survey was conducted in Saudi Arabia from January 2022 to April 2022 to determine the knowledge and competence among PT interns toward VR.

Subjects

A total of 233 PT interns were selected via convenience sampling from 19 universities across Saudi Arabia. Convenience sampling was used because it enables the selection of participants who are readily accessible and available at the time of the study. In this case, the recruitment process involved contacting PT students from multiple physical therapy institutions across Saudi Arabia, who were currently in their internship phase. The interns were approached through their clinical coordinators or program heads to ensure that all eligible candidates were reached. Inclusion criteria were PT students aged 20–29 years currently serving as interns in Saudi universities. Exclusion criteria were PT students not actively engaged in clinical internship (e.g., those who had completed their internship or were on leave) (Fig. 1).

Figure 1 Flow chart of number of universities, invitation and participant’s response rate.

All participants reviewed and signed a written informed consent form at the beginning of the survey. If the participant agreed to participate in the study, they could mark “yes”; if not, they could mark “no,” and the survey would end. Survey distribution channels (email via internship coordinators and WhatsApp professional groups), a three-week open period with two reminder emails, mandatory-response settings in Google Forms, and automatic export to Excel prior to analysis. This study was approved by the institutional review board at Majmaah University and followed the Declaration of Helsinki guidelines (MUREC-f an .6/CONI-2,022/18-4). Reporting of this observational study followed the Strengthening the Reporting of OBservational studies in Epidemiology (STROBE) statement for cross-sectional studies and the CHERRIES checklist for internet-based surveys. The completed checklists are provided as Supplementary Material.

Sample size estimation

The sample size for this study was calculated using the Raosoft sample size calculator (Raosoft, Inc.) based on a total population of 590 PT interns from universities in Saudi Arabia. The calculation incorporated a 95% confidence level and a 5% margin of error, with adjustment for finite population size. Based on these parameters, the minimum required sample size was 233 participants, which was deemed adequate to ensure representativeness of the target population.

Instrument development

The questionnaire used in this study was developed through a structured Delphi process to ensure content validity and clarity (Mengual-Andrés, Roig-Vila & Mira, 2016). The process involved three rounds of consultation with a panel of experts comprising faculty members, practicing physical therapists, and experienced researchers. In the first round, the panel reviewed a draft questionnaire developed from a comprehensive literature review and the study objectives. Feedback focused on the relevance, phrasing, and comprehensiveness of the items. Subsequent rounds incorporated modifications to improve wording of the questions, eliminate redundancy, and incorporate suggestions for improved clarity, eliminate redundancy, and enhance alignment with study goals. For instance, ambiguous items were rephrased or removed based on expert consensus. By the third round, the panel reached agreement on the final set of questions, indicating that the tool had achieved content saturation (Hasson & Keeney, 2011; Landeta, 2006).

Instrumentation

The questionnaire was developed using Google Forms in a structured format and took approximately 10–15 min to complete. The 22-item questionnaire was divided into three sections. Demographic data were collected through eight questions in Section A. In Section B of the survey instrument, competence was measured with eight dichotomous (yes/no) items evaluating participants’ exposure to patients with vestibular disorders, their perception of the role of physical therapists in vestibular care, and their self-reported confidence in evaluating and managing vestibular conditions (including identification of red-flag indicators). Knowledge was assessed in Section C: four multi-option checklist items captured which vestibular assessments and interventions participants had learned during academic training and internships, and two multiple-choice clinical vignettes evaluated applied understanding by asking respondents to identify the likely lesion type (central vs. peripheral) and the involved vestibular canal(s) based on typical patient scenarios.

Scoring rubric

Competence. Section B comprises eight dichotomous items. Each affirmative response indicating appropriate attitude, prior exposure, or self-efficacy (e.g., confidence in differential diagnosis) was scored as 1, while a negative response scored as 0, producing a competence index ranging from 0 to 8. Case-report knowledge. The final two items of Section C consisted of brief clinical case scenarios requiring (i) classification of the lesion as central or peripheral and (ii) identification of the most likely affected semicircular canal. Each correct response earned 1 point, giving a domain maximum of 2 points. Prior to data collection, the questionnaire validity and reliability were tested, and Cronbach’s alpha was calculated for the dichotomous items, yielding a value of 0.85, which indicated strong internal consistency. The questionnaire was then subjected to pilot testing with a sample of 30 PT interns with a demographic profile similar to the study population. These steps ensured that the questionnaire was both a valid and reliable tool for comprehensively capturing the study objectives.

Data and statistical analysis

All the data collected were tabulated and coded on a Microsoft Excel sheet and then transferred to IBM statistical software (JASP version 18.1.1) for data analysis. Descriptive statistics were chosen on the basis of variable type: categorical variables (e.g., sex, university affiliation) are summarized as frequencies and percentages, whereas continuous variables (e.g., age, test and treatment knowledge scores, competence scores) are reported as means ± standard deviations. For regression analyses, the assumptions of normality, multicollinearity, and homoscedasticity were met. The normality of the residuals was assessed via Q–Q plots and the Shapiro–Wilk test, which confirmed an approximately normal distribution. Multicollinearity was evaluated by calculating variance inflation factor (VIF) values for each predictor, with all values below 5. Homoscedasticity was confirmed through residual scatterplots, ensuring that variance was consistent across levels of predicted values. For four questions that included an “Other” response option, eight participants selected this choice. Four did not specify any response, and four mentioned items such as manual muscle testing, cardiopulmonary exercise test, general neurological screening, and posture assessment. Since these were heterogeneous and not related to vestibular rehabilitation, they were excluded from quantitative analyses to avoid reporting bias. The level of significance was set at 0.05.

Results

Data obtained from 233 PT interns were included in the analyses. All the participants completed all the sections of the questionnaire. The demographic data revealed that most PT interns aged between 20 and 24 years were female (n = 154, 66.4%); for a full demographic analysis, see Table 1. Additionally, to interpret the level of knowledge and competence, scores were categorized as low (<33% of the maximum possible score), moderate (33%–66%), or high (>66%) based on the proportion of the mean score relative to the maximum achievable score. A total of 233 valid responses were analyzed, with no missing data. The mean score for case report knowledge was 0.58 out of 2 (29%), indicating a low level of understanding. The competence score, which reflects overall self-reported confidence, had a mean of 5.35 out of 8 (67%), reflecting a high level of perceived competence. The academic test assessing knowledge of essential vestibular rehabilitation (VR) test maneuvers learned during academic coursework had a mean score of 4.17 out of 20 (28%), indicating low knowledge acquisition. The internship test reflecting VR test knowledge gained during clinical internship had a mean score of 2.43 out of 20 (17%), also indicating a low level of learning. Similarly, the academic treatment score, representing knowledge of core VR treatment maneuvers taught during academic years, had a mean of 2.47 out of 12 (25%), while the internship treatment score, indicating VR treatment skills learned during internship, averaged 1.69 out of 12 (19%). Both treatment-related scores reflect low levels of practical treatment knowledge.

Table 1 Demographic and educational characteristics of participating physical therapy interns (N = 233).

Variable	Category	Frequency	Percent (%)	
Age (years)	20–24	203	87.1	
25–29	30	12.9	
Gender	Male	78	33.5	
Female	155	66.5	
Professional degree	Doctor of physical therapy (DPT)	29	12.4	
Bachelor of physical therapy (BPT)	204	87.6	
Clinical internship duration	<1 month	51	21.9	
1–3 months	53	22.7	
4–6 months	49	21	
7–9 months	39	16.7	
≥10 months	41	17.6	
Formal university curriculum	No	221	94.8	
Yes	12	5.2	
Workshop attendance	No	157	67.4	
Yes	76	32.6	
Self-learning participation	No	156	67	
Yes	77	33	
Media usage for VR learning	No	88	37.8	
Yes	145	62.2	
University	Prince Sattam University	36	15.5	
Taif University	23	9.9	
Hail University	19	8.2	
Qassim University	19	8.2	
Umm Al-Qura University	19	8.2	
Imam Abdulrahman Bin Faisal University	15	6.4	
Jouf University	13	5.6	
King Khalid University	13	5.6	
Buraydah Private College	12	5.2	
Jazan University	11	4.7	
King Saud University	10	4.3	
Princess Nora bint Abdulrahman University	10	4.3	
Taibah University	9	3.9	
King Abdulaziz University	9	3.9	
Al Majmaah University	7	3.0	
Batterjee Medical College	2	0.9	
Najran University	2	0.9	
Shaqra University	2	0.9	
Tabuk University	2	0.9	

Furthermore, multiple response analyses indicate that, for learned assessment techniques during academic years, the most frequently learned test was the Timed Up and Go Test (n = 137, 58.8%), whereas during internships, the most frequently learned test was the Dix−Hallpike test (n = 51, 21.9%). For learned treatment (exercise approach) during academic years, the most frequently learned exercise approach was balance training (n = 200, 16.7%), whereas during internships, the most frequently learned exercise approach was the Vestibulo-ocular Reflex Adaptation 1 (n = 54, 23.5%) (Figs. 2 and 3). The source of knowledge indicated that a small number of participants had a formal university curriculum. In contrast, media, self-learning, and workshops were the most utilized sources for VR learning (Fig. 4). As noted in the Methods, “Other” responses were heterogeneous and therefore excluded from the quantitative summaries; their exclusion did not alter the primary outcome trends.

Figure 2 Frequency of balance and vestibular tests used by students in the academic vs. internship phase.

Figure 3 Frequency of exercise approaches and vestibular maneuvers used by students in academic vs. internship phase.

Figure 4 Various modes by which the students used to learn about vestibular rehabilitation.

To test the predictors for competence, a simultaneous-entry multiple linear regression tested whether academic vestibular-test knowledge, internship vestibular-test knowledge, academic vestibular-maneuver knowledge, internship vestibular-maneuver knowledge, case-report knowledge, age (25–28 yrs.), gender (female), workshop attendance, self-learning, media use, professional physical-therapy degree, and formal curriculum exposure predicted overall competence. The model was significant, F (12, 219) = 5.96, p < .001, explaining 24.6% of the variance (adjusted R2 = .21, RMSE = 1.78). Internship vestibular-test knowledge was the strongest unique predictor, B = 0.25, SE = 0.05, β = .32, t = 4.91, p < .001, while workshop attendance also showed a positive effect, B = 0.82, SE = 0.30, t = 2.78, p = .006.

Additionally, multiple linear regression examined whether overall competence, academic vestibular-test knowledge, internship vestibular-test knowledge, academic vestibular-maneuver knowledge, internship vestibular-maneuver knowledge, age 25–28 yrs. (vs. ≤ 24 yrs.), gender (female), possession of a professional physical-therapy degree, workshop attendance, self-learning, media use, and exposure to a formal university vestibular curriculum predicted case-report knowledge. The model was significant, F (12, 219) = 2.50, p = .004, but accounted for only 12% of the variance (adjusted R2 = .07, R = .35, RMSE = 0.74). Controlling for all covariates, workshop attendance was the sole positive predictor, B = 0.49, SE = 0.12, t = 4.09, p < .001, 95% CI [0.25–0.73], whereas holding a professional physical-therapy degree was associated with lower case-report knowledge, B = −1.06, SE = 0.53, t = −2.02, p = .045, 95% CI [−2.10, −0.02].

Discussion

This study examined how Saudi physiotherapy interns acquire and apply knowledge of vestibular-rehabilitation (VR) assessment and treatment. Although relatively few interns reported completing a dedicated VR module during their undergraduate programme, those who did demonstrated markedly stronger theoretical knowledge, underscoring the pivotal role of structured coursework. In contrast, clinical competence was shaped chiefly by experiential learning: skills gained during the internship phase most strongly differentiated interns who felt prepared to manage vestibular cases from those who did not. Continuing-education workshops further enhanced both competence and the ability to interpret case scenarios, whereas reliance on informal resources such as self-directed study or social media conferred little additional benefit. Collectively, these findings depict a learning pathway in which interns compensate for limited curricular coverage through hands-on training and professional workshops, yet still depend on formal instruction to establish a solid theoretical foundation for VR practice.

Several key factors were identified as critical in shaping physiotherapy (PT) interns’ knowledge and competence in vestibular rehabilitation (VR). During the academic years, the absence of a formally integrated VR course in the curriculum led many interns to conceptualize vestibular dysfunction primarily as a general balance impairment. Consequently, their exposure was largely limited to basic balance assessments and rudimentary treatment techniques; however, during the internship phase, PT interns gain experience with more advanced VR assessment methods, such as the Dix-Hallpike test, as well as specialized VR treatment maneuvers, including vestibulo-ocular reflex (VOR) adaptation exercises. This progression from theoretical knowledge to applied clinical practice highlights the importance of structured curricula and hands-on training in developing competence in VR. Furthermore, despite the availability of extracurricular learning resources, our results indicate that knowledge scores are primarily predicted by formal education in VR through the university curriculum, which provides a comprehensive understanding of the subject. In contrast, competence was primarily predicted by learned tests during internships, periods of clinical practice, and workshop participation, highlighting that even a minimum of 6 months of hands-on experience results in high competence among PT interns.

Additionally, the results of this study emphasize the value of formal education and training in developing knowledge and competence, as individuals with no formal education have limited knowledge in identifying commonly used tests and maneuvers for individuals with vestibular dysfunction; moreover, they do not manage to score highly on competence questions. These findings are in line with those of previous research (Sung, 2020), which reported that approximately 75% of Korean PTs had limited academic education in VR, which underscores the importance of implementing a dedicated VR educational program for PT students. Furthermore, it reinforces the need for a standardized, globally recognized vestibular medicine curriculum to ensure consistent training for health care practitioners (Van de Berg et al., 2022).

Furthermore, in Saudi Arabia, the limited knowledge of the role of PTs in managing vestibular disorders has also been noted among medical professionals (Alyahya & Kashoo, 2022), leading to the underutilization of PT expertise in assessing and treating these conditions. A previous study conducted in Germany reported that only 40% of individuals with vestibular disorders were referred for PT for assessment and treatment (Grill et al., 2014). This lack of awareness among medical professionals not only restricts the involvement of PTs in the assessment and treatment of vestibular disorders but also limits the range of therapeutic options available to patients.

Therefore, this study highlights the urgent need to implement a comprehensive academic course that integrates both assessment techniques and training maneuvers in VR. New graduates need to acquire a strong foundation in VR and pursue ongoing education to meet global professional standards. This will enable them to meet work demands and provide evidence-based and client-centered care (Alanazi & Alrwaily, 2022).

Several methodological constraints temper the generalisability of our results. A convenience, volunteer sample recruited online may have systematically excluded interns stationed in remote facilities or lacking reliable internet access, thereby introducing selection bias. The participant pool was heavily weighted toward Bachelor-level interns, with only a small cadre enrolled in the newer Doctor of Physical Therapy pathway. In addition, the cross-sectional design captures knowledge and confidence at a single time-point, limiting causal inference and obscuring developmental trajectories across clinical rotations. Although our questionnaire demonstrated good internal consistency, its self-report nature and exclusive online delivery render the data vulnerable to recall error, social-desirability bias, and coverage bias against those with limited digital literacy. Finally, all respondents were educated within Saudi Arabia, so curricular and clinical differences elsewhere may constrain external validity. A small number of “Other” responses (n = 8) were excluded from the analyses because they were either unspecified or reflected heterogeneous techniques unrelated to vestibular rehabilitation (e.g., general neurological screening, posture assessment). While this exclusion could represent a minor source of bias, the limited number of cases makes it unlikely to affect the overall findings. Future research employing probability sampling, balanced recruitment across degree pathways, longitudinal follow-up, and objective competence assessments is needed to produce a more comprehensive and transferable evidence base.

Conclusions

The study concluded that limitations in the PT curriculum may have contributed to the insufficient knowledge and competence regarding VR among PT interns. These findings underscore the need to enhance VR education by implementing educational and training programs in the undergraduate curriculum.

Supplemental Information

Supplemental Information 1 Raw Data

Raw and processed data and coding used for statistical analysis in the study. It includes demographic information, test scores, and survey responses from physical therapy interns regarding their knowledge and competence in vestibular rehabilitation.

Supplemental Information 2 Research Questionnaire

Supplemental Information 3 Checklist for Reporting Results of Internet E-Surveys (CHERRIES)

Supplemental Information 4 STROBE checklist

Additional Information and Declarations

Competing Interests

Author Contributions

Human Ethics

Data Availability

Faizan Kashoo is an Academic Editor for PeerJ.

Danah Alyahya conceived and designed the experiments, authored or reviewed drafts of the article, and approved the final draft.

Arwa Alharbi conceived and designed the experiments, performed the experiments, authored or reviewed drafts of the article, and approved the final draft.

Faizan Kashoo analyzed the data, prepared figures and/or tables, authored or reviewed drafts of the article, and approved the final draft.

Shaikha Alsaheli performed the experiments, prepared figures and/or tables, and approved the final draft.

Faisal Mubarak AlMubarak conceived and designed the experiments, authored or reviewed drafts of the article, and approved the final draft.

Rehab Aljuhni analyzed the data, authored or reviewed drafts of the article, and approved the final draft.

The following information was supplied relating to ethical approvals (i.e., approving body and any reference numbers):

The research had ethical approval from Majmaah University Ethics Number: MUREC-f an .6/CONI-2,022/18-4.

The following information was supplied regarding data availability:

The raw data is available in the Supplemental Files.

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
