# Peer review of "Knowledge and competence in vestibular rehabilitation: a cross-sectional study of physical therapy interns"

_PeerJ, doi:10.7717/peerj.20213_

## Round 0.1 · original submission · Major Revisions

· Academic Editor

Major Revisions

·

Basic reporting

The manuscript is commendably written, showcasing clear and unambiguous language throughout. The authors demonstrate a high level of professionalism in their writing style, which enhances the readability of the paper.

The literature references are extensive and pertinent, providing a solid background and context for the research topic. This contextualization effectively frames the study within the existing body of knowledge, making a compelling case for its significance.

The article structure is professional and well-organized, with logical flow and appropriate use of figures and tables that effectively illustrate key findings. Additionally, the inclusion of raw data enhances the transparency and reproducibility of the research, which is essential in scientific discourse.

The results presented are relevant to the stated hypotheses and provide clear insights, showcasing the authors' ability to connect their findings to the broader research questions. Overall, this manuscript represents a thorough and well-executed study that contributes meaningfully to the field.

Experimental design

The manuscript presents a well-structured study addressing an important and timely research question. The authors have effectively identified a significant gap in the existing literature and provide a comprehensive overview of how their research contributes to this area. The methodology is robust, and the details provided allow for replication, which is commendable.

Furthermore, the results are clearly presented, and the interpretations are insightful, contributing to the overall understanding of the topic. The authors also acknowledge the limitations of their study, which adds to the integrity of the research. Overall, this study is a valuable addition to the field and will surely stimulate further research.

Validity of the findings

The article structure is professional and well-organized, with logical flow and appropriate use of figures and tables that effectively illustrate key findings. Additionally, the inclusion of raw data enhances the transparency and reproducibility of the research, which is essential in scientific discourse.

The results presented are relevant to the stated hypotheses and provide clear insights, showcasing the authors' ability to connect their findings to the broader research questions. Overall, this manuscript represents a thorough and well-executed study that contributes meaningfully to the field.

Additional comments

good paper and accepted for publication

Reviewer 2 ·

Basic reporting

1.1. The English language should be improved throughout the manuscript, particularly in vocabulary selection and grammatical errors. Please check.
1.2. I suggest using MeSH terms as keywords.
1.3. The outcome measures are unclear. A Likert scale is reported in the Methods section, followed by averages and measures of variability in the Results section; however, it is unclear what these values correspond to. Is a score of 4.172 a good score? Please clarify this aspect.
1.4. Figure 3 is unclear. A prevalence is given in the text, but a correlation chart (Cross-tabulation) is reported. Please replace the graph with an appropriate one (e.g., a bar chart).
1.5. In Data and Statistics Analysis, the reporting needs to be improved. I would report the choice between frequencies and percentages, or mean and standard deviation, based on the nature of the variable (categorical, continuous) and not the construct it measures.
1.6. Table 1. The categorization of Clinical Internship Start Time has many overlapping time windows (i.e., less than 9 months includes all the previous labels). I suggest rephrasing as follows: < 1 month, between 1 and 3 months, between 3 and 6 months, etc.
1.7. Table 1. I suggest ordering the outcomes of each variable from most prevalent to least prevalent for better understanding.
1.8. There are several typos in the caption of Table 1.

Experimental design

2.1. It seems that no reporting checklist has been used (e.g., CHERRIES, CROSS, STROBE, etc.). Please specify the reporting guidelines and adjust the text structure accordingly.
2.2. Is there a prospective protocol registration?
2.3. Line 92. Please provide some details about convenience sampling and describe how the recruitment was conducted.
2.4. Please be cautious with the exclusion criteria, which should not be the opposite of the inclusion criteria. (E.g., if PT interns must be in their internship phase for less than one year, then they couldn’t have already completed their internship).
2.5. Lines 125-131. Likert-scale items are reported. However, I couldn’t find a Likert-scale question in the survey. Could you clarify, please?
2.6. Line 149. “descriptive statistics provided in mean ± three standard deviations”. What does three standard deviations mean in this context?
2.7. In the Results section, the regression model used is not reported. Please report the variable included in the model used.
2.8. Lines 178-179. If R² = 0.118, why did the model account for 24.8% of the variance? Could you please clarify this further?

Validity of the findings

3.1. The entire dataset is reported in a clear format; however, it seems that some data have been excluded from the reporting (i.e., respondents aged >28 and all responses to the open-ended questions). It is appropriate to avoid outcome reporting bias and include all information in the manuscript. Additionally, 233 PT interns are cited in the results, whereas 232 PT interns are reported in Table 1.

3.2. The regression analysis highlights something that may seem obvious: formal academic training has a significant impact on knowledge, while internships and workshops have a significant impact on competence. However, quantifying this expected relationship could make sense. Does it make sense to also explore relationships with other variables, such as the start time of a clinical internship or the professional degree? Did you investigate other relationships?

3.3. Lines 230-231. The second limitation did not emerge in the manuscript since no correlation analyses on professional degrees have been reported.

Additional comments

4.1. There is a problem with the references: around 16 references included in the bibliography were not cited in the text. Could you check, please?
4.2.Given the topic discussed, there is a lack of recent and relevant studies. I would add more solid references to support the manuscript.
4.3. For example, “The […] (APTA) recognized vestibular rehabilitation (VR) as a specialized area within neurology” has been referenced using the Kwon & Ko (2017) paper, which reported this in the background, not as a finding. Please use appropriate referencing.
4.4. Dougherty et al., 2020 is a book about vestibular dysfunction. I can’t find support on early VR referral effectiveness in this reference, nor the crucial role of VR in managing nausea, vomiting, etc. Please use highly relevant references from the current literature.
4.5. Lines 170-171: Balance training is not a maneuver, but more a strategy/approach/exercise
4.6. Lines 199-200. There's a typo; it’s Dix-Hallpike (please check also in other parts of the manuscript). Vestibulo-ocular reflex adaptation 1 is not a maneuver but an exercise.

·

Basic reporting

-

Experimental design

In the Materials and Methods section, the participant flowchart is not provided as per the STROBE guidelines. Additionally, the number of institutions across Saudi Arabia from which participants were recruited is not specified. The survey was conducted online via Google Forms (as provided in the supplementary material), but a more detailed description of the survey process should be included.

Validity of the findings

-

Reviewer 4 ·

Basic reporting

-

Experimental design

-

Validity of the findings

-

Additional comments

As a significant strength, this manuscript aimed to examine the knowledge and competence of Saudi physical therapist (PT) interns in VR.
The manuscript has weaknesses, including methodological issues that should be clarified.
In summary, the manuscript is really interesting and innovative, but there are several points that should be clarified. Therefore, my peer review is a major revision in this phase. I wait for the revised version of the manuscript.

AREAS OF AMELIORATION

ABSTRACT:
-Please adopt only MeSH terms as keywords.

TABLES
-Please report all the abbreviations in full in a legend.

GENERAL
-Please report ALL the abbreviations in full in the whole manuscript.
- Please revise the English

INTRODUCTION
-The background and rationale part should be developed more. I suggest improving it accordingly. Please describe what is known and what is missing, and emphasise the importance of your study in the field.
-Add the hypothesis.

METHODS
-Reporting: a cross-sectional study must adhere to the international guidelines (STROBE). Please follow it and consider all the items. I also expect more details on the methods and the completed checklist to be loaded as supplementary. Moreover, you adopted an online administration; thus, you must cite and use the CHERRIES guidelines. What I expect is to follow these guidelines, cite them, fill them, and add them as a supplementary file.

RESULTS:
-findings: please divide the results into sections following STROBE and CHERRIE

DISCUSSION
-I suggest that the authors discuss the main findings by comparing them with the existing literature in other fields, suggesting implications, and analysing strengths and limitations.

---

## Round 0.2 · Minor Revisions

· Academic Editor

Minor Revisions

Please pay particular attention to the reporting and use of appropriate checklists as noted by Reviewer 1.

**Language Note:** The review process has identified that the English language must be improved. PeerJ can provide language editing services - please contact us at [email protected] for pricing (be sure to provide your manuscript number and title). Alternatively, you should make your own arrangements to improve the language quality and provide details in your response letter. – PeerJ Staff

Reviewer 2 ·

Basic reporting

I appreciate and acknowledge the authors' effort in improving their English and giving more details about statistical analyses.

Experimental design

I couldn’t see the corrections made after the errors noted in Table 1 (i.e., overlapping time windows, frequencies order, caption typos, total number of PT included, etc). Could you check, please?

You now report adherence to STROBE and CHERRIES (online survey). However, there is no CHERRIES checklist attached. Instead, a CHEERS checklist has been provided, but it is a reporting guidance on Health Economics Research that is inadequate for your type of study.

I couldn’t find the correction reported on the exclusion/inclusion criteria redundancy.

Validity of the findings

Concerning avoiding outcome reporting bias, the qualitative responses exclusion was referring to a few respondents who selected the last response (“Other”) in the three questions in section C (see the Dataset in Supplemental files).

Additional comments

Please correct some typos in Figure 2 and Table 1 captions.

Reviewer 4 ·

Basic reporting

-

Experimental design

-

Validity of the findings

-

Additional comments

Please check the English again.

---

## Round 0.3 · accepted · Accept

· Academic Editor

Accept

Thank you for addressing the remaining concerns of the reviewers. The manuscript is now ready for publication.

·

Basic reporting

No comment

Experimental design

I appreciate the revisions made in the revised manuscript. The concerns I raised regarding the experimental design have been satisfactorily addressed. Specifically, the inclusion of the participant flowchart in line with the STROBE guidelines, the clarification of the number of institutions across Saudi Arabia involved in participant recruitment, and the expanded description of the survey process conducted via Google Forms have all strengthened the clarity and transparency of the methodology. I have no further concerns regarding these points.

Validity of the findings

No comment

Reviewer 4 ·

Basic reporting

See below

Experimental design

See below

Validity of the findings

See below

Additional comments

Dear authors,
you have improved the manuscript.
However, you need to add the reference for STROBE and CHERRIES before the acceptance.